# The Possible Role of Anti- and Protumor-Infiltrating Lymphocytes in Pathologic Complete Response in Early Breast Cancer Patients Treated with Neoadjuvant Systemic Therapy

**DOI:** 10.3390/cancers15194794

**Published:** 2023-09-29

**Authors:** Klara Geršak, Blaž Matija Geršak, Barbara Gazić, Andreja Klevišar Ivančič, Primož Drev, Nina Ružić Gorenjec, Cvetka Grašič Kuhar

**Affiliations:** 1Faculty of Medicine, University of Ljubljana, Vrazov Trg 2, 1000 Ljubljana, Sloveniabgazic@onko-i.si (B.G.); aklevisar@onko-i.si (A.K.I.); cgrasic@onko-i.si (C.G.K.); 2Division of Medical Oncology, Institute of Oncology Ljubljana, Zaloška Cesta 2, 1000 Ljubljana, Slovenia; 3Department of Pathology, Institute of Oncology Ljubljana, Zaloška Cesta 2, 1000 Ljubljana, Slovenia; pdrev@onko-i.si; 4Institute for Biostatistics and Medical Informatics, Faculty of Medicine, University of Ljubljana, Vrazov Trg 2, 1000 Ljubljana, Slovenia; nina.ruzic.gorenjec@mf.uni-lj.si

**Keywords:** breast cancer, pathologic complete response, tumor-infiltrating lymphocytes, neoadjuvant systemic therapy, CD8 antigen, forkhead box P3, programmed cell death 1 receptor, chemokine CXCL13

## Abstract

**Simple Summary:**

Understanding the role of tumor-infiltrating lymphocytes (TILs) and their phenotype in pathologic complete response (pCR) is gaining increasing attention in cancer research. It is believed that the behavior of cancer cells is affected by the tumor microenvironment, dominated by either pro- or antitumor immune cells. In our prospective study, patients with early breast cancer started treatment with systemic therapy, followed by surgery. We assessed the probability of pCR in relation to TIL density and the proportion of two (presumably) antitumor biomarkers (CD8+ and CXCL13+) and two (presumably) protumor biomarkers (PD-1+ and FOXP3+) in TILs. In addition to confirming previous findings that a higher number of TIL correlates with a higher probability of pCR, our results paradoxically showed that a higher proportion of presumably protumor TILs could be favorable for a higher probability of pCR when considering each biomarker individually (if all four biomarkers are considered together, we cannot draw any conclusions on the direction of association between protumor TILs and pCR). Future decisions regarding neoadjuvant or adjuvant therapy may be influenced by TIL density and TIL subtypes.

**Abstract:**

The tumor microenvironment, composed of pro- and antitumor immune cells, affects cancer cell behavior. We aimed to evaluate whether tumor-infiltrating lymphocyte (TIL) density and TIL subtypes in core biopsies at the diagnosis of breast cancer patients could predict a pathologic complete response (pCR; ypT0/is ypN0) from neoadjuvant systemic therapy (NST). The TIL subtypes were determined based on the proportions of presumably antitumor (CD8+, CXCL13+) and protumor (PD-1+, FOXP3+) immune cells. A prospective, noninterventional study, including 171 participants undergoing NST, was performed. The median TIL density for the entire cohort was 10% (IQR: 3.5–23.8), and 59 (35%) patients achieved pCR. TIL density was positively associated with pCR (univariately and multivariably). In the multivariable logistic regression model, TIL density was an independent predictor of pCR (*p* = 0.012, OR 1.27; 95% CI 1.05–1.54) when controlled for age (*p* = 0.232), Ki-67 (*p* = 0.001), node-negative status (*p* = 0.024), and HER2+/triple negative vs. luminal B-like subtype (*p* < 0.001). In our sample, higher proportions of PD-1+ TILs and FOXP3+ TILs were associated with a higher probability of pCR but the association was not statistically significant and we could not make any conclusions on the direction of associations in the model with all four biomarkers. In the exploratory multivariable analysis, we showed that only higher CD8+ TILs were associated with pCR. In conclusion, TIL density and its subtypes are associated with pCR.

## 1. Introduction

Breast cancer is the most common type of cancer and one of the leading causes of cancer-related mortality in women globally [1]. Five-year survival rates vary dramatically by the stage at presentation (99% in stages 0–IA and 27% only in stage IV) [2]. In the last decade, the management of breast cancer has shifted from primary surgical treatment to neoadjuvant systemic treatment (NST) in highly proliferative subtypes (human epidermal growth factor receptor 2 (HER2) positive, triple negative (TN), and some luminal B-like) [3,4,5]. This concept assumes that NST eradicates potential microscopic metastases and that achieving a pathologic complete response (pCR) confers improved long-term survival. In patients without pCR achievement, the outcome could be further improved by tailoring the post-neoadjuvant treatment [6].

The adaptive immune response has an important role in cancer growth modulation through the mechanisms of avoiding immune destruction of cancer cells and promoting inflammation [7]. Most notably, a high density of tumor-infiltrating lymphocytes (TILs) has been linked with improved outcomes in colon, ovarian, lung, and breast cancer [8,9,10,11,12]. In early breast cancer, patients with high TIL density had a higher probability of pCR after NST [12], irrespective of breast cancer subtype [13]. However, only in TN and HER2-positive subtypes did a higher TIL density contribute to prolonged progression-free survival (PFS) and overall survival (OS), while in the luminal B-like subtype this was not the case [13,14]. Stromal TIL consists of T cells, B cells, and natural killer cells in various proportions; in breast cancer, typically 75%, 20%, and 5%, respectively. The TIL subtypes further dictate the entire TIL action, thus steering cancer progression toward a favorable clinical outcome, more or less [15,16,17]. In-depth research on T cells in TILs in breast cancer patients revealed that CD8+ T-cell infiltration has been associated with a favorable clinical outcome; however, cancer cells could develop several resistance mechanisms against CD8+ T-cell antitumor activity [18,19,20,21,22,23,24]. They can secrete immunosuppressive cytokines (interleukin 6 (IL-6), IL-17, and tumor growth factor beta (TGF-β)) into the tumor microenvironment, which has been associated with a poor clinical outcome [25,26]. Moreover, these cytokines also increase the levels of tumor-associated macrophages, regulatory T cells and myeloid-derived suppressor cells in the tumor microenvironment, which further limit CD8+ T-cell tumor infiltration and activity [8,25,27,28,29,30]. This, coupled with the upregulation of programmed death ligand 1 (PD-L1) gene expression, further contributes to immunosuppression [31]. Programmed death receptor 1 (PD-1) is a cellular receptor expressed on antigen-presenting T cells. Binding with its ligands PD-L1 or PD-L2, which are normally expressed on antigen-presenting cells and aberrantly on tumor cells, leads to an inhibition of the effector functions of T cells. PD-1+ TILs were found frequently in HER2+ and basal-like tumors (part of the TN subtype). In the basal and luminal B-like subtypes, PD-1+ TILs were associated with a worse prognosis [32,33,34]; however, in some studies, it was significantly correlated with higher TIL expression and an improved probability of pCR [34,35].

Regulatory T cells in TILs have been associated with a worse prognosis, irrespective of the breast cancer subtype [8,36,37,38]. Regulatory T-lymphocytes are a subpopulation of CD4+ T-lymphocytes with the immune phenotype CD4+ CD25+ forkhead box P3 (FOXP3+). In a healthy tissue environment, they suppress and modulate immune responses to prevent autoimmune reactions. In breast carcinogenesis, the fraction of FOXP3+ T-lymphocytes increases considerably with the progression from healthy breast tissue to ductal carcinoma in situ and eventually to invasive ductal carcinoma [39,40]. In the breast cancer microenvironment, FOXP3+ lymphocytes were strongly associated with both lower relapse-free survival (RFS) and OS [41,42,43]. FOXP3 expression in the cytoplasm of tumor cells was of no prognostic significance; however, a high infiltration of FOXP3+ lymphocytes accompanied by a cytoplasmic FOXP3+ tumor was the most detrimental phenotype [37]. Other reports, however, depicted that CD4+ regulatory T cells seemed to be capable of inhibiting invasive breast cancer development from preinvasive breast cancer by suppressing protumorigenic T helper 2 cell responses [44]. Yeong et al. showed that a combination of high densities of FOXP3+, CD8+ T cells, and CD20+ B cells is associated with a favorable prognosis in the TN subtype [42,45]. 

The above-noted conflicting findings clearly show that when evaluating the percentage of FOXP3+ TILs, the percentage of CD8+ TILs should also be taken into consideration. Specifically, it is highly important to describe the ratio between the different subsets of TIL components to better understand the nature of tumor microenvironment regulation in terms of its pro- or antitumor effect [46,47].

Gene expression profiling was used to examine TILs in primary breast cancer. This showed that both CD4+ T-lymphocytes and follicular T-helper cells expressed CXCL-13. While their immune functions are currently less well understood, researchers believe that CXCL-13-expressing T-helper cells play a significant role in attracting immune cells to the tumor microenvironment, regulating antigen-specific B-cell responses, and promoting the formation of tertiary lymphoid structures, thus generating germinal centers for local memory B-cell differentiation at the tumor location [48,49,50]. Studies on the expression of CXCL13 genes showed that CXCL13 significantly correlated with improved disease-free survival [15]. One study showed that CXCL13 expression was a significant predictor of pCR [48].

However, prospective studies evaluating the contribution of different pro- and antitumor TILs to pCR are extremely scarce. The goal of this study was to analyze TIL density and TIL subtypes in core biopsies of breast cancer patients before NST and investigate their possible predictive role for pCR when treated with NST in a whole cohort and according to the breast cancer subtypes. To that end, TIL subtypes were assessed as the proportion of (presumably) antitumor (CD8+, CXCL13+) and (presumably) protumor (PD-1+, FOXP3+) immune cells in TIL specimens. Our first hypothesis was that patients with a high TIL density would have a higher probability of pCR after NST than those with a low TIL density. The second hypothesis was that higher CD8+ TILs, higher CXCL13+ TILs, lower PD-1+ TILs, and lower FOXP3+ TILs are associated with a higher probability of pCR. We also investigated TILs and their subtypes within different molecular subtypes of breast cancer and examined their association with other clinicopathologic parameters. 

## 2. Materials and Methods

### 2.1. Patients and Treatment

We conducted a prospective, noninterventional study in patients with early breast cancer who were treated with NST, followed by breast and axillary surgery. All patients were treated at the Institute of Oncology Ljubljana. Inclusion criteria were females older than 18 years, diagnosed with early breast cancer of stages IIA (cT2N0M0) to IIIB (cT4a-cN0-1M0), luminal B-like, HER2+-luminal, HER2+-non-luminal, and TN subtype, regardless of menopausal status, and intended for NST treatment.

Exclusion criteria included the luminal A-like subtype (progesterone receptors (PR) > 20% and/or Ki-67 ≤ 10%), metastatic disease, patients not understanding Slovenian, and ineligibility for NST. Before inclusion in the study, all patients provided informed consent. The Slovenian Medical Ethics Commission approved the study (No. 0120-133/2017-2, dated 8 June 2017). All procedures were carried out in accordance with the Helsinki Declaration and Good Clinical Practice.

During the diagnostic and staging procedures of breast cancer, patients underwent a clinical examination, laboratory blood analysis, bilateral mammography, breast ultrasound and ultrasound-guided core needle biopsy, radiopaque clip insertion in the breast tumor, ultrasound of tumor-side axillary and supraclavicular lymph nodes, fine-needle aspiration of any suspicious lymph node, and a breast MRI scan (unless mastectomy was planned). For metastatic disease exclusion, a contrast computer tomography scan of the thoracic and abdominal organs and bone scintigraphy was performed.

Patients were treated with NST according to European Society for Medical Oncology (ESMO) recommendations at the time [3,4]. Briefly, chemotherapy consisted of sequential therapy with anthracyclines and taxanes (four cycles of doxorubicin or epirubicin/cyclophosphamide followed by 12 cycles of weekly paclitaxel or 3–4 cycles of 5-fluorouracil/epirubicin/cyclophosphamide followed by 3–4 cycles of docetaxel). At the attending oncologist’s discretion, a dose-dense regimen was delivered. Similarly, in cases of contraindications, anthracycline-free schemas were used. Patients with HER2+ breast cancer tumors received anti-HER-2 therapy alongside taxane administration [3,4]. Breast surgery took place 3–6 weeks after the end of NST. Adjuvant radiation, endocrine and anti-HER2 therapy, and post-neoadjuvant capecitabine in TNBC were applied according to guidelines [3,4,51].

#### Endpoints

The primary endpoint was the probability of pCR after NST, defined as the absence of invasive carcinoma in the breast (in situ carcinoma could be present) and in the axilla (ypT0/is ypN0) [52]. As the outcome was a nominal variable, the term ‘association’ was used to describe the relationship with other variables, where the direction of association with the probability of achieving pCR was stated.

### 2.2. Pathohistological Examinations

On the pretreatment paraffin-embedded tumor block (core biopsy specimen), standard tumor characteristics (pathohistological subtype, grade, expression of estrogen receptors (ER) and PR, HER2 status, Ki-67) were performed. Patients were regarded as HER2+ if immunohistochemistry (IHC) was 3+; in the case of IHC 2+, fluorescent in situ histochemistry was performed, and a ratio ≥2 was regarded as HER2+. Patients were regarded as luminal B-like if ER ≥ 1%, PR ≤ 20%, Ki-67 > 10%, and HER2-. In the case of HER2+, ER 0%, and PR 0% results, patients were regarded as HER2+-non-luminal, and in the case of HER2+, ER ≥ 1%, and/or PR ≥ 1% they were regarded as HER2+-luminal. The TN subtype was defined as ER 0%, PR 0%, and HER2- [3,4]. For TIL density evaluation, one representative hematoxylin eosin (HE) stained tissue slice was used. TIL density was evaluated by two independent blinded pathologists who subspecialized in breast cancer pathology, and the average of both TIL assessments was taken for the analysis. TIL density evaluation was performed in accordance with the guidelines of the International TIL Working Group 2014 and the International Immuno-Oncology Biomarker Working Group 2017 [12,53,54]. Briefly, all cells in intratumor stroma that exhibited the characteristics of mononuclear immune cells were classified as TIL cells. TIL density was expressed as the percentage of the intratumor stroma surface area occupied by TIL and was expressed using a numerical variable ranging from 0% to 100%. TIL density was also dichotomized into a low TIL density group (0–59%) and a high TIL density group (60–100%), according to Denkert et al. [12]. For TIL phenotyping, additional tissue sections were stained separately for CD8-, CXCL13-, FOXP3-, and PD-1-positive cells. Each group of positive lymphocytes was expressed as the percentage of the intratumor stromal surface area occupied by CD8+ lymphocytes, CXCL13+ lymphocytes, PD-1+ lymphocytes, or FOXP3+ lymphocytes and was expressed as a numerical variable ranging from 0% to 100% of the total TILs. A pathologist assessed and counted the percentage of positive lymphocytes in the intratumoral stroma at 400× magnification, using the average cell percentage from four visual fields to determine the target cell percentage in each sample.

### 2.3. Assay Methods 

An immunohistochemical examination of FOXP3, CD8, PD-1, and CXCL13 was performed on 2–4 µm thick fresh formalin paraffin-embedded tissue sections dried at 56 °C for 2 h using a fully automated IHC staining system Ventana Benchmark Ultra (Ventana ROCHE Inc., Tucson, AZ, USA).

CD8, FOXP3, and PD1 epitopes were retrieved with heat-induced epitope retrieval (HIER) using Cell Conditioning Solution 1 (Ventana ROCHE Inc. Tucson, AZ, USA; cat. No. 950-124) for 88 min at 100 °C, while the CXCL13 epitope was retrieved with proteolysis using Protease I (Ventana ROCHE Inc., Tucson, AZ, USA; cat. No. 760-2018) for 8 min at 37 °C.

Retrieved epitopes were detected using monoclonal antibodies directed against CD8 (DAKO Agilent Technologies Inc. Santa Clara, CA, USA, cat. No. M7103; clone C8/144B; diluted at 1:100), FOXP3 (Epitomics, Rocklin, CA, USA, cat. No. AC0304RUO; clone EP340; diluted at 1:200), PD1 (CellMarque, Rocklin, CA, USA; cat. No. 315 M; clone MRQ22; diluted at 1:800), and CXCL13 (R&D Systems, Minneapolis, MN, USA; cat. No. MAB8012; clone 53602; diluted at 1:25).

All primary antibodies were incubated on board for 60 min at 37 °C and visualized using the three-step multimer detection system OptiView DAB IHC Detection Kit (Ventana ROCHE Inc., Tucson, AZ, USA; cat. No. 760-700) according to the manufacturer’s instructions. Visualization of CXCL13 and PD1 was enhanced using an OptiView Amplification Kit (Ventana ROCHE Inc., Tucson, AZ, USA cat. No. 760 099).

The normal appendix was used as a control for the four antigens. Optimal staining reactions were considered as follows: CD8—strong distinct membranous staining in subepithelial and interfollicular cytotoxic T cells, FOXP3—moderate intensity distinct nuclear staining in regulatory T cells, PD1—strong distinct membranous staining in the germinal center associated helper T cells and CXCL13—strong distinct cytoplasmic staining in the germinal center associated follicular helper T cells and follicular dendritic cells. All other cells must stain negatively for these antigens.

The analytical sensitivity and specificity of IHC protocols were assessed on multiple normal tissues and tumors using tissue microarrays. The CD8 protocol was validated in 17 normal tissues and 40 tumors. Of the two T-cell lymphomas, both were positive (sensitivity 100%), while the other 38 tumors were negative (specificity 100%). The FOXP3 protocol was validated in 45 normal tissues and 42 tumors, as in this study, FOXP3 IHC investigation was only used to assess the proportion of TREGs among TILs, and the analytical specificity and sensitivity were not calculated. The PD1 protocol was validated in 19 normal tissues and 45 tumors. Of the seven tested angioimmunoblastic T lymphomas, five were positive (sensitivity 71%), while the other 38 tumors were negative (specificity 100%). The CXCL13 protocol was validated in 22 normal tissues and 92 tumors. Of the 11 tested angioimmunoblastic T lymphomas, 11 were positive (sensitivity 100%), while all other 82 tumors were negative (specificity 100%). The CD8 protocol has passed external quality assessment in several programs (UK NEQAS, NordiQC and LabQuality) with excellent scores, while thus far, there is no option to test PD1, FOXP3, and CXCL13 protocols. 

### 2.4. Statistical Analysis 

#### 2.4.1. Sample Size

With the planned sample of 180 patients, a chi-square test with a significance level of 0.05 achieves 80% power to reject the null hypothesis of no association between TIL (high ≥60% or low) and pCR (yes or no) if Cohen’s effect size W is at least 0.21. This effect size corresponds to the difference in the probability of pCR between the high and low TIL groups of at least 29% if we assume that in the population, 30% achieve pCR (based on [55]) and 12% belong to the high TIL group (based on [12]). When analyzing TIL as a numerical variable, we are able to detect even smaller effect sizes.

#### 2.4.2. Data Analysis

Numerical variables are presented as medians and interquartile ranges (IQRs), and categorical variables are presented as frequencies and percentages. In addition to the descriptive statistics, the association of each patient or tumor characteristic with pCR was evaluated by univariate logistic regression, where *p* values were adjusted using Holm’s method to control the family-wise error rate. The distribution of TILs was presented graphically using boxplots. The difference between TIL measurements by two pathologists is presented by a Bland–Altman plot (Appendix A). On bar plots for the proportion of pCR in separate TIL groups, 95% Clopper–Pearson’s exact confidence intervals (CIs) are presented.

The association between TIL and pCR was assessed in a multivariable logistic model, where we controlled for age, nodal stage, Ki-67 (numerical), and cancer subtypes (two groups based on different biology: combined HER2+ luminal, HER2+ non-luminal, and TN vs. luminal B-like subtype). The model meets the requirement for a sufficient number of events per variable with 59 events (achieved pCR) and five estimated coefficients (the selection of independent variables was predetermined and based on background knowledge). In exploratory analysis (Appendix A), separate multivariable models were fitted to explore the association between pCR and separate TIL subtypes (CD8+, CXCL13+, FOXP3+, PD-1+) using the same independent variables as in the main model for TIL. In addition, a multivariable model for pCR with all four TIL subtypes and age, nodal stage, Ki-67, and cancer subtypes was fitted to further explore the association between pCR and TIL subtypes (although it does not meet the sufficient number of events per variable). All logistic models were fitted using Firth’s bias reduction method, and odds ratios (ORs) were reported together with 95% CIs. The discriminative ability of the multivariable logistic models was estimated by means of ROC (receiver operating characteristic) curve analysis reporting the area under the ROC curve (AUC).

In the exploratory analysis (Appendix A), the association between TILs and various clinical and histopathologic characteristics (tumor grade and size, categorized Ki-67, nodal status, breast cancer subtype) was evaluated using the Mann–Whitney or Kruskal–Wallis test for numerical TILs and Fisher’s exact test for dichotomized TILs, where *p* values were adjusted using Holm’s method to control the family-wise error rate.

An (adjusted) *p* value of less than 0.05 was considered statistically significant. The analysis was performed with the help of relevant open-source libraries in Python 3.8.5. and SPSS, IBM Co., v. 22.

## 3. Results

### 3.1. Patient Population

Between February 2018 and March 2021, we enrolled 268 patients. Of those, 171 met the inclusion criteria, and all were included in the final analysis (Figure 1).

The median age of the patients was 48 years (IQR 41.7–57.4). Of them, 91 (53%) were of the luminal B-like subtype, 32 (19%) had HER2+ luminal, 16 (9%) had HER2+ non-luminal, and 32 (19%) had the TN subtype. The patients’ tumor and clinical characteristics are presented in Table 1 (second column). Briefly, 161 patients (96%) had invasive carcinoma of no special type. The majority were grade 3 (124, 73%); the median Ki-67 was 30% (IQR 20–50); the median tumor size was 30.5 mm (IQR 23.3–40), and 125 (73%) of patients were node positive. A pCR was achieved in 59 (35%) patients.

The results of univariate analysis for a comparison of the clinicopathologic characteristics of patients who achieved pCR to their counterparts are also presented in Table 1. There was a statistically significant difference in the probability of pCR among breast cancer subtypes (adjusted *p* < 0.001). The highest probability of pCR was in patients with the HER2+-non-luminal subtype (88%), and the lowest was in patients with the luminal B-like subtype (18%). Grade 3 tumors and tumors with high Ki-67 had a higher probability of pCR (both adjusted *p* < 0.001). In our sample, patients with negative lymph nodes had a higher probability of pCR, but this difference could not be generalized to the population (adjusted *p* = 0.224).

### 3.2. Association of TIL and Its Subtypes with pCR 

#### 3.2.1. TIL Density and Its Association with pCR

The median TIL density in all patients was 10% (IQR 3.5–23.8). The median interobserver absolute difference was 5.0 (IQR 0.5–14.5) (Appendix A), and interobserver variability was low at low TIL density but high at high TIL density (Appendix A). The HER2+-non-luminal subtype had the highest median value of TILs (22.5%), followed by the TN subtype (16.3%). The distribution of TIL (expressed as a numerical value) among the breast cancer subtypes is presented in Appendix A. Based on the predetermined cut-off of 60%, 161 (94%) tumors belonged to the low TIL group and 10 (6%) belonged to the high TIL group. In this high TIL group, 5 patients (31%) belonged to luminal B-like, 4 (27%) to TN, and 1 (7%) to the HER2+-non-luminal subtype. Representative IHC slides of low and high TIL are presented in Figure 2(A1,A2).

TIL categorization into low and high categories did not show a significant association with either of the clinicopathological characteristics. However, numerical TILs were significantly positively associated with tumor grade and Ki-67 (both adjusted *p* = 0.002) (Appendix A). For instance, median TIL in G3 tumors was 12.5%, compared to only 5% in G2 tumors. Similarly, tumors with high Ki-67 (>40%) had a median TIL of 17.5% compared to 5% in low Ki-67 (10–20%).

Importantly, the numerical TIL density was associated with pCR (univariate logistic model, adjusted *p* = 0.003) (Table 1). The association between TILs and pCR is graphically presented in Figure 3. A higher probability of pCR (60% vs. 33%) was also observed when comparing the high TIL group with the low TIL group (cut-off of 60%), but the difference was not statistically significant (*p* = 0.088, adjusted *p* = 0.351). In Figure 4, we present an explorative analysis of the proportion of pCR according to TIL density in separate breast cancer subtypes (not formally tested due to a low number of patients achieving pCR (event) in subtype groups). The 0–10% TIL group had substantially lower pCR than the other groups. In all subtypes, the probability of pCR was 100% when TIL was 75–100% (however, small sample sizes in these groups of patients imply large confidence intervals). In the HER2+-non-luminal subtype, the probability of pCR was high regardless of the TIL density (again with large confidence intervals).

In our multivariable logistic model, numerical TIL concentration was shown to be an independent factor for the prediction of pCR when controlled for age, Ki-67, lymph nodes (positive/negative), and molecular subtypes (dichotomized into luminal B/other) with *p* = 0.012 (Table 2). The model discriminated the data well with AUC = 0.817. With an increase in TIL density of 10%, the odds for pCR increased by 27%. Patients with clinically negative axillary lymph nodes had 2.5 times higher odds of pCR than node-positive patients. Patients with HER2+/TN breast cancer subtypes had 5.1 times higher odds of achieving pCR than those with the luminal B-like subtype. For patients with an increase in Ki-67 of 10%, the odds for pCR increased by 38%.

#### 3.2.2. TIL Subtypes and Their Association with pCR

Representative IHC slides of TIL subtypes are presented in Figure 2 (low expression: B2–E2; high expression: B1–E1). An investigation of TIL subtypes revealed that CD8+ TILs were the most prevalent component of TILs with a median of 40.0% (IQR 30.0–50.0), whereas other TIL subtypes were present in lower percentages: the median percentage of CXCL13+ TILs was 1% (IQR 0.5–2.0), that of PD1+ TILs was 2% (IQR 0.0–5.0), and that of FOXP3+ TILs was 4% (IQR 1.0–7.0) (Table 1, Figure 5). Thirty-one percent, 21%, and 5% of tumors did not express any PD-1, CXCL13, or FOXP3, respectively. Figure 6a–d show the distribution of TIL subtypes among the breast cancer subtypes. Briefly, all breast cancer subtypes had similarly high CD8+ TILs (median 40–45%). There was no difference in the expression of CXCL13+ TILs among the breast cancer subtypes. On the other hand, (presumably) the protumor TIL subtypes PD-1+ TILs and FOXP3+ TILs showed a tendency to be more prevalent in the TN and HER2+-non-luminal subtypes (Figure 6, Appendix A), but this could not be generalized to the population (PD-1+: *p* = 0.062, adjusted *p* = 1.0; FOXP3+: *p* = 0.027, adjusted *p* = 0.598). The associations between TIL subtypes and clinicopathological characteristics are shown in Appendix A. There was a statistically significant association between PD-1+ TILs and grade (adjusted *p* = 0.015) and Ki-67 (adjusted *p* = 0.024), and between FOXP3+ TILs and grade (adjusted *p* = 0.012), although these differences were not meaningful. We also found an association of CXCL13+ TILs with the tumor stage (adjusted *p* = 0.051).

In the exploratory multivariable logistic model, only CD8+ TIL (numerical) was significantly associated with pCR when controlled for age, Ki-67, lymph nodes (positive/negative), and molecular subtypes (dichotomized as luminal B/other), *p* = 0.045 (Appendix A). Increasing CD8+ TILs by 10% increases the odds for pCR by 31%. The model discriminated the data well with AUC = 0.807. Similar exploratory multivariable logistic models for CXCL13+ TILs, PD-1+ TILs, and FOXP3+ TILs did not show an association with pCR (Appendix A).

Surprisingly, PD-1+ TILs and FOXP3+ TILs were positively associated with pCR in our sample, although the association was not significant in univariate (PD-1+: *p* = 0.053, adjusted *p* = 0.325; FOXP3+: *p* = 0.051, adjusted *p* = 0.325) or multivariable analysis (PD-1+ *p* = 0.131, FOXP3+ *p* = 0.193). Furthermore, in an additional multivariable model (Appendix A) where all TIL subtypes were included together with age, Ki-67, lymph nodes, and molecular subtypes, we could not make any conclusions on the direction of associations due to very wide confidence intervals (PD-1+: OR with 95% CI 1.31 (0.66–2.66), *p* = 0.429; FOXP3+: OR with 95% CI 1.21 (0.39–3.63), *p* = 0.708). This model should be interpreted with caution as it includes too many variables for the number of patients achieving pCR (events).

## 4. Discussion

In our prospective noninterventional study, performed in early breast cancer patients undergoing NST, we investigated the possible role of pretreatment TIL density and TIL subtypes in achieving pCR. Our data clearly confirmed the first hypothesis—that patients with a high TIL density would have a higher probability of pCR after NST than those with a low TIL density.

On the other hand, our second hypothesis, that higher CD8+ TILs, higher CXCL13+ TILs, lower PD-1+ TILs, and lower FOXP3+ TILs are associated with a higher probability of pCR, was partially confirmed: only high CD8+ TILs were associated with pCR in the multivariable model (exploratory analysis). Surprisingly, in our sample, a higher probability of pCR was associated with a higher percentage of PD-1+ TILs and FOXP3+ TILs but these associations were not statistically significant in multivariable models or in univariate analyses. Furthermore, we could not make any conclusions on the direction of associations when the four biomarkers were considered together in one model due to confidence intervals that were too wide. Other independent predictive factors for a higher probability of pCR were the HER2+ non-luminal/TN subtype, high Ki-67, and grade (univariate analyses, adjusted for multiple comparisons).

In this study, we added TIL density evaluation to the routine histopathological analysis of the tumor sample analysis before treatment. TIL density in our cohort was positively associated with grade and Ki-67 (Appendix A). The median TILs were 12.5% and 17.5% in G3 and high Ki-67 (Ki-67 > 40%) tumors, respectively. A similar correlation of high TILs with high Ki-67 and grade was reported in some other studies [56,57,58].

In our sample, patients with the HER2+-non-luminal subtype had the highest median percentage of TILs (22.5%) among the four different subtypes, but this difference could not be generalized to the population (adjusted *p* = 0.085) (Appendix A).

In our sample, only 10 patients had high TIL (>60%) and six of them (60%) achieved pCR compared to 33% pCR in the low TIL group, but the association between dichotomized TIL (cut-off 60%) and pCR was not statistically significant. The reason could be the small number of patients in the high TIL group or TIL cut-off value that was too high. 

Similar but significant findings, however, were reported in the GeparDuo cohort, where a subgroup of lymphocyte-predominant breast cancer (>60% of either stromal or intratumoral lymphocytes) had a probability of pCR of 41.7% in comparison with 9.3% in the other group [12,59]. Additionally, in the GeparDuo and GeparTrio cohorts, the odds ratio for pCR increased with the extent of TILs [60]. Tumors without any TILs had a probability of pCR of 7.2%, while those with lymphocyte-predominant breast cancer had a probability of pCR of 40% [12,53,61]. It should be noted that the probability of pCR was 12.8% and 17% in the GepardDuo and GepardTrio cohorts, respectively, whereas it was 35% in our study. We used a TIL cut-off of 60% because it was the standard used by researchers in large neo- or adjuvant trials at the time our study was designed and used a cut-off between 50% and 60% [59,62,63,64,65,66,67,68,69]. Numerous other studies, performed in the meantime, showed substantially lower cut-offs for high TIL to be appropriate to distinguish between patients who benefited from pCR prediction. The cut-off, however, depends on the breast cancer subtype [63]. Recently, the St. Gallen consensus meeting for the TN subtype recommended a cut-off of 50%. This cut-off was based on relapse-free survival (RFS) data, not pCR data, and emphasized a very favorable outcome in small TN tumors without adjuvant chemotherapy treatment [70]. For HER2+ subtypes, even lower cut-offs (5–40%) are reported to be prognostic for pCR [62,63]. Our sample size of 171 patients with 59 achieving pCR (events) was too small to allow for a cut-off determination.

In our study, numerical TILs were significantly associated with pCR in univariate analysis (Table 1) and were revealed as an independent factor for pCR in a multivariable model (Table 2). Increasing TIL by 10% increased the odds for pCR by 27% (Table 2). Figure 3 indicates that a dichotomized TIL is not suitable for determination of the association with pCR. Figure 4 suggests that all subtypes of breast cancer were linked to higher percentages of pCR when the TIL density was high in our sample, but our sample size did not permit an additional inferential subgroup analysis. Similarly, in the GeparSixto trial, it was shown that increased levels of stromal TILs predict pCR in the whole cohort and in the TN and HER2+ cohorts [65,71]. Our multivariable model confirmed the HER2+/TN subtype and node-negative status as additional independent predictive factors for a higher probability of pCR. These findings are in accordance with reports of others [12,62,63,71,72,73,74,75].

It is now increasingly clear that the tumor microenvironment is composed of heterogeneous tumor cells and the host’s endogenous stroma, which together undergo changes as the disease progresses. Importantly, stromal cells are becoming recognized as key players in the development of the tumor microenvironment, metastasis, immune infiltration and inflammation, and resistance to chemotherapeutic agents [76,77,78].

Our investigation revealed that CD8+ TILs were the most prevalent among the studied TILs, accounting for a median of 40% of the total TILs (Table 1, Figure 5). Additionally, all breast cancer subtypes had similarly high CD8+ TILs (median 40–45%). In contrast, presumably protumor TIL subtypes (PD-1+ TILs and FOXP3+ TILs) tended to be more prevalent in the TN and HER2+ non-luminal subtypes in our sample (Figure 6, Appendix A). 

In our sample, numerical CD8+ TILs did not show any association with grade, Ki-67, tumor size, nodal stage, or subtypes (Appendix A). However, the number of CD8+ TILs was significantly associated with pCR in a multivariable model (exploratory analysis, Appendix A). The odds for pCR increased by 31% with an increase in CD8+ TILs of 10%. 

From other reports, there is some evidence that the possible role of CD8+ T cells is greater in hormone receptor-negative breast cancers. In a study evaluating 1854 breast cancer samples, Baker et al. showed improved disease specific survival in patients with high CD8+ TILs in ER-negative tumors only. In contrast, in low-grade ER-positive tumors, a high number of CD8+ TIL was associated with inferior outcomes [79]. CD8+ infiltrates were seen in 60% of TNBCs. In concordance with our results, CD8+ TILs were found to be an independent predictive factor for pCR [80,81].

CXCL13+ TILs represented only a median of 1% of total TILs and were similarly expressed in all cancer subtypes. We only found an association with the tumor stage. In two other studies, CXCL13 expression was associated with higher tumor grade (grades 2–3), positive nodes, ER-negative status, longer metastasis-free survival, and a stronger prognostic effect in HER2-positive breast cancer [49,82]. In the third study, a four-gene signature that includes CXCL13 was predictive of the extent of lymphocytic infiltration after NST in TNBC. The increase in the value of these signatures was associated with better distant relapse-free survival, adding novel prognostic information for this aggressive breast cancer subtype [83].

In our patients, PD-1+ TIL expression was associated with high tumor grade and high Ki-67, similar to the findings of others [34,35,84]. Importantly, we observed that a higher percentage of pCR is associated with higher PD-1+ TILs, but this association was not significant when generalized to the population, and we could not draw any conclusions when we controlled for the other three biomarkers—CD8+, CXCL13+, and FOXP3+ (Appendix A). A significant association of PD-1 expression with higher TIL scores and pCR in early breast cancer, however, was demonstrated by Denkert et al. [65]. This suggests that PD-1 expression may be a useful biomarker for predicting the response to NST in breast cancer patients. Further studies are needed to confirm these findings and determine the potential clinical implications.

In our sample, TN and HER2+-non-luminal subtypes had a slightly higher median expression of FOXP3+ TILs than other subtypes, but this could not be generalized to the population. Loi et al. reported the highest percentage of FOXP3+ cells in the TN subtype [61], but the median percentage in that study was 70%, which is substantially higher than that in our study (4%). FOXP3+ TILs are known to disrupt antitumor immunity by suppressing the effector functions of various immune cells and have been implicated in the escape of cancer cells from immunosurveillance [85,86]. Increased levels of FOXP3+ cells were strongly associated with an increased risk of early and late relapse, lower RFS and OS [12], and a lower percentage of pCR [19]. In our study, on the contrary, FOXP3+ TILs were positively associated with pCR in our sample when considered individually (not statistically significant), while we could not make any conclusions on the direction of the association when controlled for other biomarkers—CD8+, CXCL13+, and PD-1+. Some other authors, especially in recent years, also found a positive association of FOXP3+ TILs with pCR [43,87,88]. Sun et al. reported subtype-specific improvement of pCR (OR: 1.20, 95% CI 1.02–1.40) and OS for the HER2+ subtype of breast cancer [43]. The possible mechanism underlying this paradoxical observation could be the higher efficacy of chemotherapeutic treatments in a highly immunosuppressive environment. Namely, it has been reported that some chemotherapeutic agents, particularly cyclophosphamide, can inhibit FOXP3+ cells. Chemotherapy might be more effective in tumors with high levels of FOXP3+ TILs, and this in turn could facilitate tumor attack by CD8+ TILs and help to achieve pCR [43]. Furthermore, tumors with a high CD8+/FOXP3+ ratio were more sensitive to anthracycline and taxane chemotherapy [22] and had higher pCR, especially the TN and HER2+ subtypes [22,89]. 

To summarize, our data suggest that an evaluation of TIL subtypes by IHC could add additional predictive information for pCR beyond TIL density alone. More sophisticated and expensive techniques, such as software-guided immune cell evaluation, could improve the accuracy of TIL subtype evaluation. However, we have to take into consideration that these techniques may suffer from preanalytical and interpretative inconsistencies, which could impair their reproducibility and widespread application in daily practice. It is very important to clearly define each immune cell subset studied. For example, Fillion et al. reported CD4 expression not only in T cells but also in a subpopulation of monocytes, causing bias and discrepancies in results and interpretation between studies [90]. 

This study has some limitations. First, core needle biopsy samples could not be fully representative of the whole tumor’s heterogeneity; however, it is used everywhere as a standard test to decide on the type of first treatment. Second, the visual method of TIL scoring is prone to intra- and interobserver variability. In our case, interobserver variability was low at low TIL density but high at high TIL density (Appendix A). In the case of high discordance, a consensus decision between pathologists should be made. Third, immunophenotyping of TILs includes only a few types of immune cells and does not take into consideration other subtypes of lymphocytes, natural killer cells, and cytokines or nonimmune cells, such as macrophages, neutrophils, and myeloid-derived stem cells. Furthermore, we could validate the data with another technique (fluorescence staining or gene expression).

There are several strengths to our study. First, our study is very valuable because it was a prospective study that included a substantial fraction of the patients treated with NST in a three-year period. Second, for our pathology department, the study was very important because, within the study procedures, we initiated the systematic scoring of TIL in core biopsies of breast cancer samples, and the method is now ready to be implemented in routine clinical work. TIL density is evaluated using HE-stained sections only, which makes it practical for routine use. Additionally, preoperative breast cancer tumor board meetings were set up at the beginning of our study, resulting in better decision-making about the treatment sequence for all newly diagnosed breast cancer patients. We believe that the incorporation of TIL density measurements can be easily incorporated into routine pathology, especially in low-to-middle-income countries, but some run-in time is needed, as was the case with the Ki-67 method.

Further implication of our work: There is still much work to be conducted in the field of TIL biomarkers, especially in the course of immunotherapy treatment, which has been approved for the treatment of the TN subtype of breast cancer. The link between pCR, when treated with immunotherapy, and TIL subtypes should be investigated. Such research could pave the way for personalized treatment plans based on TIL biomarkers, leading to improved outcomes for breast cancer patients. Additionally, future studies could explore the potential of TIL biomarkers in predicting responses to other forms of cancer treatment beyond immunotherapy. In the future, machine learning techniques may help us further standardize TIL evaluation and find important immune phenotypes.

In summary, TIL scoring is a new biomarker that could be used in clinical practice. It remains open as to whether PD-1+ TILs and FOXP3+ TILs are linked with a higher or lower response to chemotherapeutics used in the neoadjuvant treatment of breast cancer.

## 5. Conclusions

TIL scoring is a new biomarker associated with achieving pCR with NST independently of known pathohistological factors that could be easily incorporated into routine pathologic evaluation. It remains open as to whether TIL subtypes linked with presumably protumor characteristics, such as PD-1+ TILs and FOXP3+ TILs, are linked with a higher or lower response to chemotherapeutics used in the NST of breast cancer. 

## Figures and Tables

**Figure 1 cancers-15-04794-f001:**
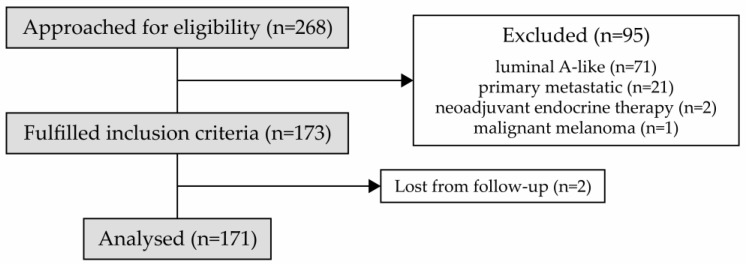
CONSORT flow diagram for the study.

**Figure 2 cancers-15-04794-f002:**
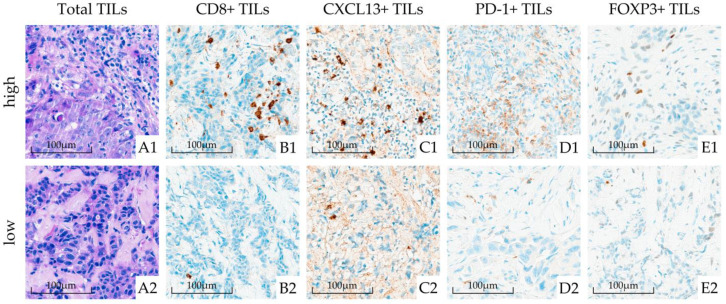
Histopathological slides of patients with total tumor-infiltrating lymphocytes (TIL) (high on (**A1**) and low on (**A2**)) and TILs subtypes (CD8+ TILs, CXCL13+ TILs, PD-1+ TILs, and FOXP3+ TILs) (high on (**B1**,**C1**,**D1**,**E1**); low on (**B2**,**C2**,**D2**,**E2**)).

**Figure 3 cancers-15-04794-f003:**
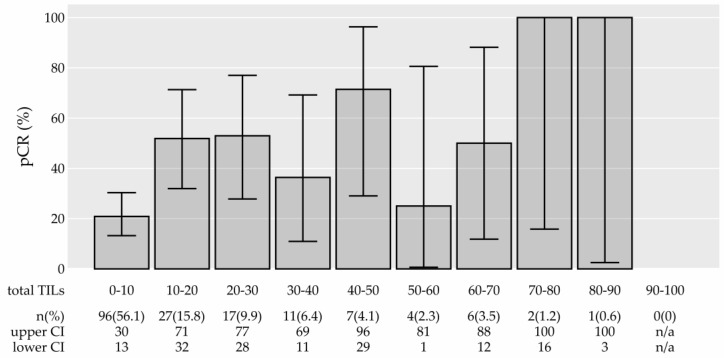
The proportion of pathologic complete response (pCR) in 10% tumor-infiltrating lymphocyte (TIL) groups, together with 95% confidence intervals (CI). The association between TIL and pCR was formally tested via a univariate logistic model (*p* < 0.001, adjusted *p* = 0.003, see Table 1) and a multivariable logistic model (*p* = 0.012, see Table 2), where controlled for age, Ki-67, lymph nodes (positive/negative), and molecular subtypes (dichotomized into luminal B like/other). Abbreviations: n/a: not applicable.

**Figure 4 cancers-15-04794-f004:**
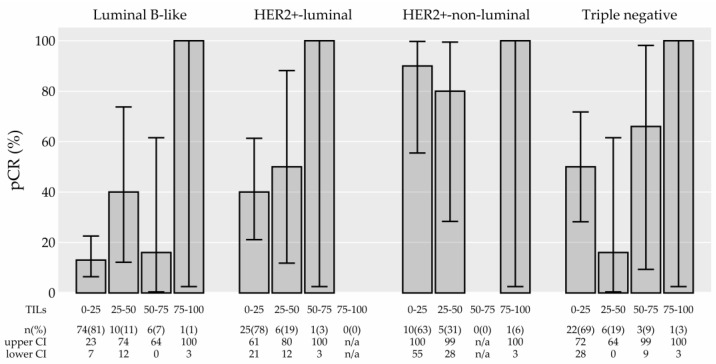
The proportion of pathologic complete response (pCR) in 25% tumor-infiltrating lymphocyte (TIL) groups in different molecular subtypes, together with 95% confidence intervals (CI). The association between TIL and pCR was not formally tested in separate molecular subtypes due to a low number of patients achieving pCR (event) in subtype groups. Abbreviations: HER-2: human epidermal growth factor receptor 2; n/a: not applicable.

**Figure 5 cancers-15-04794-f005:**
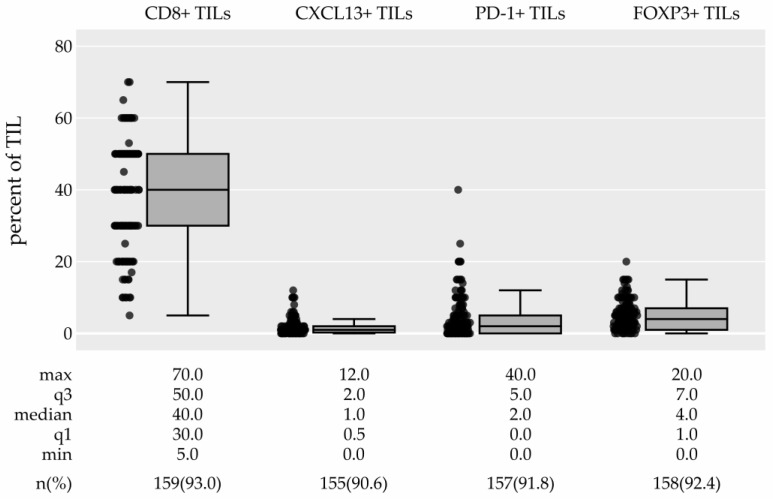
Percentages of CD8+, CXCL13+, PD-1+, and FOXP3+ lymphocytes in tumor-infiltrating lymphocytes (TILs), boxplots. Abbreviations: min: minimum; max: maximum; q3: 3rd quartile; q1: 1st quartile; n: number.

**Figure 6 cancers-15-04794-f006:**
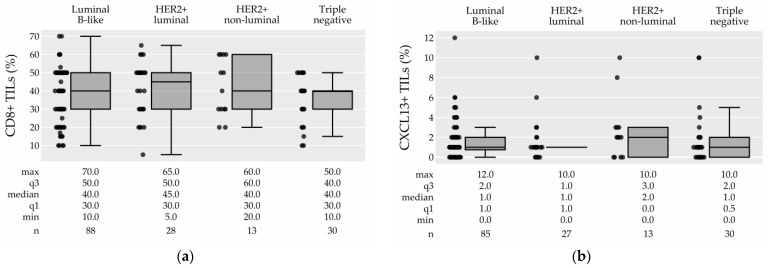
The distribution of CD8+ TILs (**a**), CXCL13+ TILs (**b**), PD-1+ TILs (**c**), and FOXP3+ TILs (**d**) among the breast cancer subtypes, boxplots. The difference in separate TIL subtypes between breast cancer subtypes was statistically significant for FOXP3+ TILs only (*p* = 0.027), but this could not be generalized to the population (adjusted *p* = 0.598), see Appendix A. Abbreviations: min: minimum; max: maximum; q3: 3rd quartile; q1: 1st quartile; TILs: tumor-infiltrating lymphocytes; HER-2: human epidermal growth factor receptor 2; n: number.

**Table 1 cancers-15-04794-t001:** Univariate analysis of patient characteristics and tumor features, and their association with pathologic complete response (pCR).

Characteristic (Number of Missing Values)	All Patients (n = 171)	Patients with pCR (n = 59)	Patients without pCR (n = 112)	OR (95% CI)	*p*-Value	Adjusted *p*-Value ^†^
Age median (IQR) (0)	48.4 (41.7–57.4)	46.4 (39.4–56.0)	49.0 (44.7–57.5)	0.97 (0.94–1.00)	0.046	0.325
Histology n (%) (2)					0.014	0.122
IBC of no special type	161 (96.0%)	58 (36.0%)	103 (64.0%)	7.35 (0.84–964.63)	0.077	
ILC	6 (4.0%)	0 (0.0%)	6 (100.0%)	ref.	ref.	
Other	2 (1.0%)	0 (0.0%)	2 (100.0%)	2.60 (0.01–553.14)	0.642	
Grade n (%) (1)						**<0.001**
Grade 3	124 (73.0%)	54 (44.0%)	70 (56.0%)	5.83 (2.41–16.89)	<0.001	
Grade 2	46 (27.0%)	5 (11.0%)	41 (89.0%)	ref.	ref.	
Molecular subtype n (%) (0)					<0.001	**<0.001**
Luminal B-like	91 (53.0%)	16 (18.0%)	75 (82.0%)	ref.	ref.	
Her2+-luminal	32 (19.0%)	14 (44.0%)	18 (56.0%)	3.59 (1.50–8.65)	0.004	
Her2+-non-luminal	16 (9.0%)	14 (88.0%)	2 (12.0%)	26.54 (7.23–144.62)	<0.001	
Triple negative	32 (19.0%)	15 (47.0%)	17 (53.0%)	4.05 (1.71–9.77)	0.002	
Ki-67 median (IQR) (1)	30.0 (20.0–50.0)	50.0 (30.0–60.0)	30.0 (20.0–40.0)	1.04 (1.02–1.05)	<0.001	**<0.001**
Ki-67 n (%) (1)					<0.001	**<0.001**
10–20%	46 (27.0%)	3 (7.0%)	43 (93.0%)	ref.	ref.	
20–40%	66 (39.0%)	25 (38.0%)	41 (62.0%)	7.64 (2.58–30.11)	<0.001	
>40%	58 (34.0%)	31 (53.0%)	27 (47.0%)	14.24 (4.78–56.53)	<0.001	
Tumor stage n (%) (15)					0.325	0.650
T1 (≤20 mm)	25 (16.0%)	10 (40.0%)	15 (60.0%)	ref.	ref.	
T2 (>20≤50)	113 (72.0%)	39 (35.0%)	74 (65.0%)	0.78 (0.33–1.92)	0.584	
T3 (>50 mm)	18 (12.0%)	6 (33.0%)	12 (67.0%)	0.77 (0.22–2.60)	0.670	
Lymph nodes n (%) (0)						0.224
Positive	125 (73.0%)	37 (30.0%)	88 (70.0%)	ref.	ref.	
Negative	46 (27.0%)	22 (48.0%)	24 (52.0%)	2.17 (1.09–4.33)	0.028	
Total TILs median (IQR) (0)	10.0 (3.5–23.8)	17.5 (7.5–35.0)	5.0 (3.0–17.5)	1.03 (1.01–1.05)	<0.001	**0.003**
Total TILs n (%) (0)						0.351
Low (<60%)	161 (94.0%)	53 (33.0%)	108 (67.0%)	ref.	ref.	
High (≥60%)	10 (6.0%)	6 (60.0%)	4 (40.0%)	2.93 (0.85–10.99)	0.088	
CD8+ TILs median (IQR) (12)	40.0 (30.0–50.0)	40.0 (30.0–50.0)	40.0 (27.5–50.0)	1.02 (1.00–1.04)	0.122	0.366
CXCL13+ TILs median (IQR) (16)	1.0 (0.5–2.0)	1.0 (1.0–2.0)	1.0 (0.0–2.0)	1.03 (0.89–1.19)	0.673	0.673
PD-1+ TILs median (IQR) (14)	2.0 (0.0–5.0)	3.0 (1.0–8.0)	1.0 (0.0–5.0)	1.05 (1.00–1.12)	0.053	0.325
FOXP3+ TILs median (IQR) (13)	4.0 (1.0–7.0)	5.0 (2.0–7.0)	3.0 (1.0–6.0)	1.08 (1.00–1.18)	0.051	0.325

^†^ *p* values are adjusted for multiple comparisons between groups. Univariate analysis was performed. Statistically significant adjusted *p*-values are presented in bold. Abbreviations: IBC: invasive breast carcinoma; ILC: invasive lobular carcinoma; TIL: tumor-infiltrating lymphocytes; pCR: pathologic complete response; OR: odds ratio; HER-2: human epidermal growth factor receptor 2; CI: confidence interval; ref.: reference; IQR: interquartile range; n: number.

**Table 2 cancers-15-04794-t002:** Multivariable logistic model of prognostic factors for pathological complete response (AUC = 0.817, log likelihood ratio test *p* < 0.001).

Characteristic	OR (95% CI)	*p*-Value
Total TILs (per 10%)	1.27 (1.05–1.54)	**0.012**
Age	0.98 (0.95–1.01)	0.232
Ki-67 (per 10%)	1.38 (1.13–1.70)	**0.001**
Negative lymph nodes	2.51 (1.13–5.69)	**0.024**
HER2+-luminal, HER2+-non-luminal, TN vs. lumB	5.11 (2.43–11.36)	**<0.001**

Statistically significant *p*-values are presented in bold. TIL: tumor-infiltrating lymphocytes; TN: triple negative subtype; lumB: luminal B subtype; HER-2: human epidermal growth factor receptor 2; OR: odds ratio; CI: confidence interval; AUC: area under the curve.

## Data Availability

Data will be available after considering the aim of further use.

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
