# Peer review of "The Possible Role of Anti- and Protumor-Infiltrating Lymphocytes in Pathologic Complete Response in Early Breast Cancer Patients Treated with Neoadjuvant Systemic Therapy"

_cancers, 2023, doi:10.3390/cancers15194794_

Round 1

Reviewer 1 Report

The authors present a prospective, non-interventional study of early stage breast cancer patients (n=171) and investigate the significance of tumor infiltrating lymphocytes (TILs), CD8, IGKC, FOXP3, PD-1 for pathologic complete response (pCR) after neoadjuvant therapy. They show that TIL density was positively associated with pCR in univariable and multivariable analysis. Surprisingly, higher proportions of PD-1+ TILs and FOXP3+ TILs were associated with a higher likelihood of pCR even though this association was not statistically significant. In an exploratory multivariable analysis, only CD8+ TILs were significantly associated with pCR.

The study is well conducted and well written. The results are interesting and confirm the influence of TILs. As is often seen in correlative studies, the results for additional immunohistochemically detected biomarkers are inconsistent. It may be interesting to elaborate that presumably pro- and antitumor biomarkers are often strongly correlated, which could lead to seemingly contradictory results such as the observed higher pCR in tumors with a higher proportion of FOXP3+ TILs.  In addition, the authors should report the numbers and percentages of each subgroup (i.e., TIL 0-10% and so on or luminal-B-like and so on) in Figures 3-6.

Author Response

Dear Reviewer 1.

Thank you very much for taking the time to review this manuscript.

Please find the detailed responses and the corresponding corrections highlighted in red in the attached and re-submitted files.

Yours sincerely.

Klara Geršak

Reviewer 2 Report

The manuscript explores the role of phenotyping tumor infiltrating immune cells (TIL) as a predictive marker for pathologic complete response (pCR)/ prognosis in breast cancer. While the usage of TIL phenotype for success of immunotherapy is an area of intense study in this field, this study has several concerns.

1) Technically, the authors have performed their work solely based on immunohistochemistry (IHC) using epitope retrieval assay and staining with DAB-based detection. The assay controls (positive and negative) and standardization procedure, sensitivity, specificity and other assay characteristics are not provided.

2) The IHC specimens should be scored, instead of labeling as just positive or negative and data analysis should be performed according to a score. This is important as currently, there is no difference between high frequency infiltration versus low frequency, as long as it reaches a basic cut-off.

3) In figures 3-6, the statistical significance between various cohorts is NOT shown within the figure nor in the figure legends.

4) As CD8 cells can also express PD1 and FoxP3, the current procedure cannot differentiate between double staining and single staining of immune cells. The authors should use atleast one more technique to validate their data (fluorescence staining or gene expression).

5) The authors use the word 'association' in their manuscript. Association versus correlation should be clearly defined in the statistical terms. Possibly consider using a professional statistician's help.

minor editing and proof-reading required.

Author Response

Dear Reviewer 2.

Thank you very much for taking the time to review this manuscript.

Please find the detailed responses and corresponding corrections highlighted in red in the attached and resubmitted files.

Yours sincerely.

Klara Geršak

Round 2

Reviewer 2 Report

No further comments